# Comparative Genomics Analysis Combined with Homologous Overexpression Reveals the Mechanism of Species-Specific Acid Stress Resistance in *Bifidobacterium animalis*

**DOI:** 10.3390/foods14244243

**Published:** 2025-12-10

**Authors:** Li Lin, Yanyu Zhang, Fazheng Ren, Shaoyang Ge, Yanling Hao, Zhengyuan Zhai, Ming Zhang, Shusen Li, Liang Zhao, Erna Sun

**Affiliations:** 1College of Food Science and Nutritional Engineering, China Agricultural University, Beijing 100083, China; linli@cau.edu.cn (L.L.); yanyuzhang0422@163.com (Y.Z.); zhaizy@cau.edu.cn (Z.Z.); lzhao@cau.edu.cn (L.Z.); 2Key Laboratory of Functional Dairy, Department of Nutrition and Health, China Agricultural University, Beijing 100193, China; renfazheng@263.net (F.R.); haoyl@cau.edu.cn (Y.H.); 3Research Center for Probiotics, China Agricultural University, Sanhe 065200, China; geshaoyang@foxmail.com; 4School of Food and Health, Beijing Technology and Business University, Beijing 100024, China; zhangming@th.btbu.edu.cn; 5Mengniu Hi-Tech Dairy Product Beijing Co., Ltd., Beijing 101100, China; lishusen@mengniu.cn

**Keywords:** *Bifidobacteirum animalis*, acid tolerance, acid tolerance, comparative genomics, homologous overexpression

## Abstract

Most *Bifidobacterium* species exhibit poor stress tolerance, which greatly limits their stability and industrial application. In contrast, *Bifidobacterium animalis* shows remarkable tolerance and distinct species-specific characteristics. Here, the survival rate of 17 strains from six *Bifidobacterium* species was evaluated under pH 2.5 for 2 h. All strains of *B. animalis* maintained survival rates above 60%, whereas other species fell below the detection limit. Comparative genomic analysis among *B. animalis*, *B. breve*, *B. longum*, and *B. bifidum* revealed 328 unique genes that were conserved across *B. animalis* strains but absent in the other species. Transcriptomic analysis of the *B. animalis* subsp. *lactis* A6 strain under acid stress identified ten genes that were significantly upregulated at both early and late stages of exposure. Homologous overexpression confirmed that four of these genes markedly enhanced survival under acid stress. Among them, *BAA6_RS00480*, encoding a GlsB/YeaQ/YmgE family stress response membrane protein, increased the survival rate by more than 22-fold. These findings provide new insight into the species-specific genetic mechanisms underlying the exceptional acid tolerance of *B. animalis*.

## 1. Introduction

The human gastrointestinal tract harbors *Bifidobacterium* spp. as part of its natural microbiota. Several *Bifidobacterium* species, such as *B. animalis* [1,2], *B. bifidum* [3], *B. breve* [4], and *B. longum* [5,6], are believed to support host health and are used as probiotics in the food industry. However, poor stress tolerance is one of the important factors hindering the application of *Bifidobacterium*. Given that *Bifidobacterium* is mainly used in yogurt or fermented milk in the food industry and that it faces an extremely acidic environment in the gastrointestinal tract, acid stress should be considered. Generally, most *Bifidobacterium* are sensitive to acid, and the optimal growth pH is 6–7 [7]. When the acidity of fermentation products is too high, the activity of *Bifidobacterium* is seriously diminished [8,9,10]. Therefore, elucidating the tolerance mechanisms of *Bifidobacterium* is essential for developing technologies to maintain viability and enhance the activity of probiotic strains.

*Bifidobacterium animalis* is a one-of-a-kind species and the most widely exploited species in various products. This is largely due to its outstanding tolerance to various stresses during production and storage [11,12,13], in marked contrast to the weak stress tolerance characteristic of other *Bifidobacterium* species [14,15,16,17]. Matsumoto et al. [18] reported that after 3 h at pH 3, all *B. animalis* strains showed a reduction of less than 1 (log_10_). In contrast, *B. longum*, *B. adolescentis*, and *B. pseudocatenulatum* suffered a much greater loss, with reductions exceeding 5 (log_10_). Despite the acid tolerance of *B. animalis* having been explored, many identified responses are universal in the *Bifidobacterium* genus [18]. The genes involved are widely distributed among various species and thus incapable of explaining the stark interspecies differences in acid tolerance. While strains generally respond to stress by rapidly adjusting their intracellular metabolism, the exceptional acid tolerance of *B. animalis* appears to be fundamentally determined by its genomic composition from the beginning. This suggests that the unique genes of *B. animalis* may hold the key to its superior performance.

Some studies have begun to focus on species specificity to uncover the common molecular mechanisms of a species. Notably, *Bifidobacterium* possess unique core metabolic pathways that distinguish them from other bacteria, such as the bifid shunt with its key enzyme fructose-6-phosphate phosphoketolase, which governs the characteristic fermentation profile yielding acetate and lactate [19]. It was reported that a unique gene in *B. aesculapii* species isolated from captive common marmosets (*Callithix jacchus*) empowered a species-specific metabolic ability to utilize acacia [20]. Genes encoding NADH peroxidase, catalase, and superoxide dismutase found in *B. asteroids* strains were responsible for the high oxygen tolerance of this species compared with other oxygen-sensitive strains [21,22]. However, the species-specific mechanism of acid resistance in *B. animalis* is still ambiguous.

Despite the exceptional acid tolerance of *B. animalis*, the underlying species-specific genetic determinants remain ambiguous. To achieve this, unique genes of *B. animalis* involved in the acid tolerance responses were identified by comparative genomics with transcriptomics and functionally validated their roles through homologous overexpression in the *B. animalis* subsp. *lactis* A6 strain. By focusing on species-unique gene content rather than only inducible stress responses, this work extends beyond previous studies and provides new mechanistic insight into the exceptional acid tolerance of *B. animalis*, a distinction not shared by other *Bifidobacterium* species. These results provide new strategies for improving the acid tolerance of *Bifidobacterium*, thereby facilitating more robust production processes in the fermentation and probiotic industries.

## 2. Materials and Methods

### 2.1. Strains and Preculture

*B. animalis* subsp. *lactis* A6, *B. longum* subsp. *longum* BBMN68, *B. longum* subsp. *longum* BLS, *B. longum* subsp. *longum* 7, *B. bifidum* L, and *B. adolescentis* 42 were obtained from our laboratory collection. *B. longum* subsp. *longum* NCC2705 was kindly provided by the Nestlé Research Center (Lausanne, Switzerland). *B. animalis* subsp. *lactis* BB-12 was obtained from Chr. Hansen A/S (Hørsholm, Denmark). *B. animalis* subsp. *lactis* DSM 10140 and *B. animalis* subsp. *animalis* JCM 11658 were obtained from the China General Microbiological Culture Collection Center (Beijing, China). *B. animalis* subsp. *lactis* CICC 6167, *B. animalis* subsp. *lactis* CICC 6174, *B. animalis* subsp. *lactis* CICC 21715, *B. animalis* subsp. *lactis* CICC 6250, *B. animalis* subsp. *lactis* CICC 6165, *B. longum* subsp. *infantis* CICC 6069, and *B. breve* CICC 6079 were obtained from the China Center of Industrial Culture Collection (Beijing, China).

All of the strains from frozen cells or freeze-dried cells were anaerobically cultured using Anaero-Pack Anaerok (Mitsubishi Gas Chemical, Tokyo, Japan) in de Man-Rogosa-Sharpe (MRS) medium supplemented with 0.5 g/L L-cysteine hydrochloride at 37 °C for 12 h. Before being used in experiments, the cells were precultured twice.

### 2.2. Acid Tolerance

The method was modified from Shen [23]. All of the strains were anaerobically cultured in MRS medium with or without 200 μg/mL spectinomycin at 37 °C to mid-exponential phase. Following incubation, the cells were collected by centrifugation (8000× *g* for 10 min). The collected cells were suspended in an equal volume of MRS broth, which had previously been acidified with HCl to pH 2.5, and the culture was grown anaerobically at 37 °C. After 0, 0.5, 2, 2.5, or 4 h of incubation, plate count was performed on MRS agar. The survival rate was defined as the number of viable cells after acid treatment/the number of viable cells before acid treatment. The lethal rate was defined as “1—the survival rate”.

### 2.3. Genome Sequences

A total of 47 sequenced *Bifidobacterium* genomes, i.e., 15 *B. animalis* genomes, 5 *B. bifidum* genomes, 10 *B. breve* genomes, and 17 *B. longum* genomes, were obtained from the public databases of NCBI (Appendix A).

### 2.4. Functional Genome Distribution Analysis

Functional genome distribution (FGD) analysis was performed using 47 publicly available *Bifidobacterium* genome sequences. FGD analysis evaluates the functional relatedness among microorganisms based on their predicted open reading frames (ORFs) [24]. An all-versus-all BLAST (version 2.13.0). comparison of the combined ORFeomes was conducted to evaluate inter-ORF amino acid sequence similarity. The resulting pairwise data were aggregated using the unweighted pair group method with arithmetic mean [25] and visualized as a phylogenetic tree in MEGA (version 5.10) [26]. Functional annotations, mainly clusters of orthologous groups databases (COGs), were generated using the metagenome analysis tool GhostKoALA (version 2025.02) [27].

### 2.5. Calculations and Analyses of the Gene Repertoire

Gene repertoire analyses were performed for the four species in this study according to methods that were previously used [28]. In brief, orthologous clusters were created using the Perl-script collection GET_HOMOLOGUES (https://github.com/eead-csic-compbio/get_homologues, accessed on 2 December 2025), which applied the following for identification of the coding DNA sequence into orthologous groups: -E < 1 × 10^−5^ for BLASTP searches and -C 75% minimum alignment coverage. The gene repertoire was determined using the OrthoMCL algorithm. A gene repertoire matrix was created using the script compare_clusters with settings: -d including only OrthoMCL data; -m produce intersection in gene repertoire matrix; and -s report the cloud-, shell-, softcore-genome and conserved-genome. GET_HOMOLOGUES defined these compartments empirically, as follows: conserved, genes contained in all genomes; cloud, genes present in ≤2 genomes; shell, genes present in >2 genomes and not present in >1 genome; softcore, genes not present in ≤1 genome [28]. The cloud-, shell-, softcore-, and conserved-genomes were analyzed with GhostKOALA (version 2025.02). The results were visualized with R (version 4.3.2).

### 2.6. Identification of Unique Genes in B. animalis

Identification of unique genes was performed according to methods that were previously used [28]. Identification of clade unique genes was performed using the pare_pangenome_matrix script of GET_HOMOLOGUES, with the following options: -A, a list of genomes of clade A, including the 15 *B. animalis*; -B, a list genomes of clade B, including the other 32 *Bifidobacterium* in this study; -g and -P 90, finding genes present in 90% genomes of clade A and absent in 90% genomes of clade B. Unique genes were analyzed with GhostKOALA using -l mapping unique genes into the referred genome. The results were visualized with R.

### 2.7. RNA Sequencing and qRT-PCR

Total RNA was isolated and prepared for RNA sequencing and qRT-PCR. Total RNA was isolated from acid-stressed cells and the non-treated cells using TRIzol reagent (Invitrogen, Carlsbad, CA, USA), and genomic DNA was removed using DNase I (Takara, Beijing, China). RNA quality was determined using agarose gel electrophoresis and a NanoDrop 2000 spectrophotometer (NanoDrop Technologies, Wilmington, DE, USA) and further assessed using an Agilent 2100 Bioanalyzer (Agilent Technologies, Santa Clara, CA, USA).

Strand-specific libraries were constructed using a TruSeq Stranded mRNA Sample Prep kit (Illumina, San Diego, CA, USA) according to the manufacturer’s instructions. The resulting qualified Illumina pair-end library was used for sequencing. RNA sequencing was performed using the Illumina HiSeq 4000 (pair-end, 150 bp × 2) sequencing platform. Raw sequencing reads were trimmed using SeqPrep and Sickle with default parameters. The clean reads were then mapped to the genome of strain A6 using orientation mode in Bowtie2 [29]. For gene expression estimates, a standard calculation of fragments per kilobase of exon per million mapped reads (FPKM) was performed using RNA-Seq by Expectation-Maximization [30]. Prior to differential gene expression analysis, the read counts were adjusted using normalization factors computed by the trimmed mean of M-values method. EdgeR (Version 3.24.3) was used to identify the differentially expressed genes [31]. Genes with a change in expression of more than 2.0-fold and a false discovery rate < 0.05 were considered to be significantly altered.

To measure the relative expression of target genes by qRT-PCR, cDNA was obtained by reverse transcribing total RNA with an ABM 5× All-In-One Master Mix (ABM, G592, Richmond, BC, Canada). Primers, cDNA, SYBR Green I Real-time PCR Master Mix (Takara, Beijing, China), and nuclease-free water were mixed on ice. The tubes (NEST, Beijing, China) were centrifuged at low speed for 30 s to settle the reaction volume and eliminate air bubbles. Tubes were placed onto the LightCycler 96 Real-Time PCR (Roche, Basel, Switzerland) and the cycling program was executed. The 16S rDNA gene was used as an internal reference gene. Data were processed with LightCycler^®^ 96 SW 1.1 and SPSS (Version 27.0.1). The sequences of the primers for measuring the relative expression of the 10 overexpressed genes are listed in Appendix A.

### 2.8. Construction of Overexpression Strains

The method was modified from Zuo [32]. Genomic DNA from the *B. animalis* A6 strain was isolated by using a bacterial DNA kit (DP302, Tiangen, China). The sequences of the genes *BAA6_RS02390*, *BAA6_RS02980*, *BAA6_RS03885*, *BAA6_RS05205*, *BAA6_RS06440*, *BAA6_RS06240*, *BAA6_RS00480*, *BAA6_RS06435*, *BAA6_RS02185*, and *BAA6_RS06445* were downloaded from NCBI (GenBank accession number NZ_CP010433.1). The primers for PCR were designed according to the choice of restriction sites in the pDP152 plasmid and the sequence of the target genes by Primer Premier 5.0 (Appendix A). The genes were PCR amplified and verified for the correct gene size on a 1% agarose gel. PCR products were purified by using a Cycle Pure kit D6492 (OMEGA, Norwalk, CT, USA). The pDP152 plasmid with the *B. longum* sucrase promoter and spectinomycin resistance gene Spr was a gift from the laboratory of Shangwu Chen (China Agricultural University, Beijing, China). Cultures with 1% (*v*/*v*) *E. coli* carrying the pDP152 plasmid were inoculated in Luria–Bertani medium with 50 µg/mL spectinomycin at 37 °C for 12 h with shaking. Then, 10 mL of culture solution was centrifuged at 6000× *g* for 10 min. The plasmid was extracted according to the instructions of a Plasmid Mini kit II (OMEGA, Norwalk, CT, USA). The plasmid was digested with SmaI and HindIII (Takara, Beijing, China) by using a double-digestion system. For genes BAA6_RS06435 and BAA6_RS02185, a single-digestion system was used due to adjacent digestion sites. After digestion by NsiI (NEB, Ipswich, MA, USA), the products were digested by HindIII (Takara, Beijing, China). Each gene was inserted using a DNA ligation kit (Takara, Beijing, China). Recombinant plasmids were transformed into competent *E. coli* DH5α cells (Tiangen, CB101, Beijing, China) and plated on Luria–Bertani agar containing 50 µg/mL spectinomycin. Positive colonies were selected and the successful construction of the recombinant vector was verified by colony PCR and sequencing analysis.

Subsequently, recombinant plasmids were isolated by using a Plasmid Mini kit II (OMEGA, Norwalk, CT, USA) and transformed into the B. animalis A6 strain with electroporation using a Bio-Rad Gene Pulser Xcell (Bio-Rad, Hercules, CA, USA). *B. animalis* A6 was inoculated at 1% (*v*/*v*) in MRS medium supplemented with 0.5 g/L L-cysteine hydrochloride and 171.15 g/L sucrose at 37 °C overnight until the OD600 was 0.4. The cells were collected by centrifugation (8000× *g* for 5 min) and washed with washing buffer (0.5 mol/L sucrose, 1 mmol/L triammonium citrate, pH 6.0, 4 °C). The cells were collected by repeating the above steps twice. Recombinant plasmid and receptive cells were mixed on ice and incubated for 5 min. The mixture was transferred to a shock cup pre-cooled at 4 °C for electric shock. The parameters were 1.75 kV, 25 µF, and 200 Ω. Then, the mixture was transferred to MRS medium supplemented with 0.5 g/L L-cysteine hydrochloride and 171.15 g/L sucrose and cultured at 37 °C for 3–4 h. The culture was diluted and plated on Luria–Bertani agar containing 200 µg/mL spectinomycin. Positive colonies were selected and verified by PCR and sequencing analysis. The level of expression of different genes was detected by qRT-PCR.

## 3. Results

### 3.1. Assessment of Acid Tolerance Among Different Bifidobacterium Species

To evaluate acid tolerance across *Bifidobacterium*, 17 strains from six species were examined. After 2 h at pH 2.5, the survival rate of all of the *B. animalis* strains was found to be over 60%, while no clones were detected for other *Bifidobacterium* spp. (Figure 1), confirming that the acid tolerance of *B. animalis* strains was pronounced. In contrast, the other *Bifidobacterium* spp. strains, such as *B. longum*, *B. infantis*, *B. brevis*, *B. bifidum*, and *B. adolescentis*, had weak acid tolerance. Their survival rate was under the detection limit, and the viable cells decreased in number by over 8 (log10). These results indicated that the acid tolerance level of *B. animalis* depends on species differences, rather than on strain differences.

### 3.2. Gene Repertoire Analysis of Four Bifidobacterium Species

To obtain an overview of the studied *Bifidobacterium* species, functional genome distribution (FGD) analysis was performed based on sequences of 47 genomes from *B. animalis*, *B. longum*, *B. breve*, and *B. bifidum*. FGD analysis showed a clustering of species that corresponded with their phylogenetic relationships (Figure 2). Notably, a clear division was found between *B. animalis* species and the three other *Bifidobacterium* species genomes, indicating that the functional relationship of *B. animalis* species was more distant and distinguished from that of the other species. This provided a hint about the differences in phenotypes between the *B. animalis* strains and other species strains.

Subsequently, gene repertoire analysis of four species was performed to understand the genetic characteristics. As shown in Figure 3, the size of total genes was distinguished from each other, with 1740 genes in *B. animalis*, 4124 genes in *B. longum*, 2855 genes in *B. breve*, and 2157 genes in *B. bifidum*, of which the total genes of *B. animalis* had far fewer than the others. Nevertheless, 1289, 1061, 1304, and 1362 conserved genes were determined in *B. animalis*, *B. loumg*, *B. breve*, and *B. bifidum*, respectively (Figure 3A–D). Genes were divided into three groups: cloud (present in ≤2 genomes), shell (3 ≤ *n* < N − 1 genomes), and softcore (present in ≥N − 1 genomes). N is the total number of genomes analyzed. Among the four species, the number of cloud and shell genes of *B. animalis* was the least (Figure 3A), indicating more conserved variation across strains. Furthermore, the gene repertoires were annotated by clusters of orthologous groups of proteins (COGs) (Figure 3E–H). For all of the species, fewer cloud and shell genes were distributed in “translation, ribosomal structure, and biogenesis” (COG J), “energy production and conversion” (COG C), “nucleotide transport and metabolism” (COG F), “coenzyme transport and metabolism” (COG H), “lipid transport and metabolism” (COG I), “cell cycle control, cell division, and chromosome partitioning” (COG D), “posttranslational modification, protein turnover, and chaperones” (COG O), and “intracellular trafficking, secretion, and vesicular transport” (COG U) categories, indicating that these functions were housekeeping functions for *Bifidobacterium* species. However, the other COGs showed more extensive diversity across strains within a species because they contained more cloud or shell genes. Genetic variation in these categories may be associated with the different ecological niches in which strains reside and their specific phenotype [33].

Overall, despite the difference in the composition of the gene repertoires between *B. longum*, *B. breve*, and *B. bifidum*, the distribution patterns of the COGs were remarkably similar (Figure 3F–H), especially the high proportion of the “carbohydrate transport and metabolism” (COG G) category, which is the most well-known characteristic of *Bifidobacterium*. Interestingly, the distribution pattern of *B. animalis*, which was different from that of other studied species, highlighted the distinctiveness of its genome (Figure 3E). The result was consistent with the FGD analysis. Genes of *B. animalis* were more frequently annotated to the “amino acid transport and metabolism” (COG E) and COG J categories, while they tended to be in the COG G and “replication, recombination and repair” (COG L) categories in the three other species. Even in these categories, which contained more cloud and shell genes, such as COG G and COG L, *B. animalis* showed lower diversity. The highly conserved and individualized genome sculptured the species-specific phenotype of *B. animalis*.

### 3.3. Identification of Unique Genes by Comparative Genomics

To identify determinants of the species-specific properties of high stress tolerance in *B. animalis*, a comparative analysis was performed. By using the pare_pangenome matrix script of GET_HOMOLOGUES, 328 genes conserved in *B. animalis* species but absent in the three other species were identified and named as unique genes, which accounted for 25.4% of the conversed genes in *B. animalis* (Figure 4A) (Appendix A). This provided genetic clues for the formation of its species-specific phenotypes. Among the 148 annotated genes, COG G, “inorganic ion transport and metabolism” (COG P), “cell wall/membrane/envelope biogenesis” (COG M), and COG E constituted the four most abundant categories (Figure 4B), showing the species-specific functional preference of *B. animalis*. However, about 55% (180 genes) of the truly unique genes were unannotated. The lack of sufficient knowledge on the function of these unannotated genes in *B. animalis* leaves their biological implication ambiguous.

### 3.4. Transcriptomic Analysis of B. animalis A6 Under Acid Stress

To further ascertain the involvement of these unique genes in the acid tolerance response, a transcriptomic analysis was performed on the *B. animalis* A6 strain after exposure to pH 2.5 for 30 min. The RNA sequencing data revealed that 543 genes were differentially expressed after acid treatment. Of these, 246 genes were upregulated, of which 66 were unique genes of *B. animalis*; and 297 genes were downregulated, of which 79 were unique genes (Figure 5A, Appendix A). The putative functions of differential genes were classified by COGs. Except for COG G and “energy production and conversion” (COG C) categories, the number of downregulated genes was always greater than the number of upregulated genes in the functional categories. This might imply that the strain globally diminished the level of non-essential metabolisms when it encountered stresses, so that intracellular energies were more concentrated in processes that strengthened resistance. In contrast, the expression of genes in COG G and COG C was increased due to their key role in energy production.

Regulation of the non-unique genes was similar to that of the total regulated pool (Figure 5B,C). For the unique genes, there were several different trends. The expression of all of the genes in COG E, COG P, COG I, and COG F was downregulated, while genes in “signal transduction mechanism” (COG T) were upregulated (Figure 5D). The analysis of unique genes supports excluding the general stress response in *Bifidobacterium* and instead shows a unique trend of adjusted intracellular function in *B. animalis*.

To further identify genes that play crucial roles in acid tolerance, the time of acid treatment (pH 2.5) was extended to 150 min, at which point the lethal rate reached approximately 50%. Genes significantly upregulated at both the early (30 min) and late (150 min) stages of acid stress were identified, based on their potential for sustained involvement in the stress response. Ultimately, 10 genes of A6 were identified whose expression levels increased more than two-fold at both 30 min and 150 min at pH 2.5 (Figure 5E). Among these, genes *BAA6_RS00480*, *BAA6_RS02185*, *BAA6_RS02980*, and *BAA6_RS06435* encoded a GlsB/YeaQ/YmgE stress response membrane protein, peptidase C45, GTP-binding protein, and protein-tyrosine-phosphatase, respectively, and the products of the other six genes (*BAA6_RS02390*, *BAA6_RS03885*, *BAA6_RS05205*, *BAA6_RS06240*, *BAA6_RS06440*, and *BAA6_RS06445*) were hypothetical proteins.

### 3.5. Homologous Overexpression

The effect of each target gene on acid tolerance was verified by plasmid-borne homologous overexpression constructs. Genes *BAA6_RS00480*, *BAA6_RS02185*, *BAA6_RS02390*, *BAA6_RS02980*, *BAA6_RS03885*, *BAA6_RS05205*, *BAA6_RS06240*, *BAA6_RS06435*, *BAA6_RS06440*, and *BAA6_RS06445* were successfully cloned into the pDP152 expression vector and designated as pDP152-*BAA6_RS00480* to pDP152-*BAA6_RS06445*, respectively. Following sequencing, the recombinant plasmids (containing the *B. longum* sucrase promoter) were transformed into *B. animalis* A6, yielding the recombinant strains (designated as A6-*BAA6_RS00480* to A6-*BAA6_RS06445*). A6-pDP152 harboring the empty pDP152 vector served as the control. The mRNA expression levels for each overexpressed gene in the A6 strain were analyzed by qRT-PCR. All of the plasmid-borne target genes in the recombinant strains were found to be significantly overexpressed (>2-fold) compared to the corresponding endogenous gene expression in the control strain A6-pDP152 (Figure 6A).

To investigate their acid resistance, these overexpression strains were challenged with an acid stress environment of pH 2.5 for 4 h (Figure 6B). The survival rates of the wild-type A6 strain and the A6-pDP152 strain containing empty plasmids were as low as 0.93% and 1.34%, respectively. However, strains overexpressing *BAA6_RS00480*, *BAA6_RS03885*, *BAA6_RS05205*, and *BAA6_RS06240* showed stronger acid tolerance, and their survival rates were 30.1%, 12.9%, 15.1%, and 12.8%, respectively, indicating that overexpression of these four genes significantly improved the acid tolerance of *B. animalis* A6. Genes *BAA6_RS03885*, *BAA6_RS05205*, and *BAA6_RS06240* encoded hypothetical proteins, and the lack of information on these proteins limited the comprehension of their functions. *BAA6_RS00480* was annotated as a GlsB/YeaQ/YmgE stress response membrane protein and was predicted to encode a protein with 82 amino acids. GlsB was predicted to encode an acyltransferase activity protein, which improved the defense of the cell membrane under stressful environments.

## 4. Discussion

This study confirmed the species-specific and robust acid tolerance of *B. animalis* and identified genomic features that distinguish this species from other members of the *Bifidobacterium* genus. Comparative genomic analysis of 47 genomes showed that *B. animalis* possesses a smaller but more conserved gene repertoire, together with 328 unique genes absent from *B. breve*, *B. longum* and *B. bifidum*. The response of these unique genes in the *B. animalis* A6 strain to acid was further explored by transcriptomics, and 10 upregulated genes were overexpressed in the A6 strain. The overexpression of *BAA6_RS00480*, annotated as a GlsB/YeaQ/YmgE stress response membrane protein, most effectively improved the acid tolerance of the A6 strain. This is the first study to reveal the mechanism of species-specific acid tolerance in *B. animalis*.

The overexpression of genes *BAA6_RS00480*, *BAA6_RS03885*, *BAA6_RS05205*, and *BAA6_RS06240* improved the survival rate of the A6 strain under pH 2.5 for 4 h, especially the overexpression of gene *BAA6_RS00480*, which increased the survival rate 22-fold. The distribution of these genes in other strains was further checked by NCBI BLAST. Homologs of genes *BAA6_RS03885*, *BAA6_RS05205* and *BAA6_RS06240* encoding hypothetical proteins only existed in *Bifidobacterium*, indicating that these three genes might be specific products of the evolutionary development of the *Bifidobacterium* genus. Gene *BAA6_RS00480*, encoding GlsB, a general stress protein first identified in *Enterococcus faecalis,* which was found to be involved in various stresses, such as bile tolerance [34,35]. This was the first time that homologs of GlsB from *Enterococcus faecalis* were reported to be involved in the acid tolerance of *B. amimalis*. GlsB has acyltransferase activity and is essential for the synthesis of acylated amino acids, which are important components of the cell membrane in some bacteria [36]. Some acylated amino acids have been found to improve the growth of *Rhizobium tropici* and *Burkholderia cepacia* under acidic and thermal stress [37,38]. Nevertheless, whether this protein directly modifies membrane lipids, interacts with other stress-response regulators, or influences proton homeostasis requires further experimental evidence. Therefore, the study acknowledges the lack of mechanistic validation as a limitation. Future research employing lipidomics, membrane fluidity assays, and structural biology will be necessary to precisely define the role of this protein family in acid tolerance of *B. amimalis*. Furthermore, these genes will be heterologously expressed in other stress-sensitive *Bifidobacterium* species, with the aim of empowering a broader range of strains with improved tolerance.

*Bifidobacterium* has become a classic genus in commercial probiotics due to its well-documented health benefits [39,40,41]. Industrially, *Bifidobacterium* are commonly preserved and applied in liquid, spray-dried, frozen, or lyophilized forms [42]. They are now widely incorporated into various foods and supplements, including yogurt, fermented milk beverages, cheese, probiotic solid beverages, compressed candy, and dietary supplements. Viability is a prerequisite for probiotic function [43]. It is generally accepted that a probiotic must reach a concentration of 10^6^–10^8^ CFU/mL (or per gram) in the human intestine to exert its beneficial effects effectively [44]. However, the complex stress environments encountered during the production, processing, storage, and distribution of bifidobacterial products lead to a severe decline in viable cell counts, significantly compromising product reliability and efficacy [45]. The acid-tolerance genes identified in this work may serve as genetic targets for engineering more robust strains. For instance, introducing key genes like *BAA6_RS00480* into other *Bifidobacterium* species such as *B. longum* or *B. breve* may could fundamentally improve their resilience. This strategy could enhance the stability and survival of probiotic cultures in low-pH dairy products like yogurt and fermented milk, ensuring higher viable counts reach the consumer. Furthermore, it could expand the applicability of *Bifidobacterium* into a wider range of acidic functional beverages and foods, thereby unlocking new product development opportunities in the functional food sector.

## 5. Conclusions

This work revealed the distinctive genome behind the species-specific acid resistance of *B. animalis*. Homologous overexpression strains were successfully constructed to verify their functions in the acid stress response. Gene *BAA6_RS00480* encoding the GlsB/YeaQ/YmgE stress response membrane protein increased the survival rate of the *B. animalis* A6 strain 22-fold. These results provide new evidence for the acid response mechanism of *Bifidobacterium* and are conducive to the development of research on improving acid tolerance.

## Figures and Tables

**Figure 1 foods-14-04243-f001:**
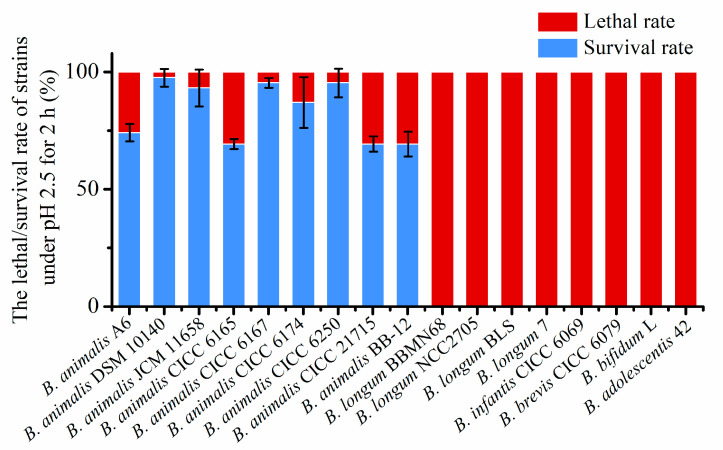
The lethal/survival rate of *Bifidobacterium* strains under pH 2.5 for 2 h.

**Figure 2 foods-14-04243-f002:**
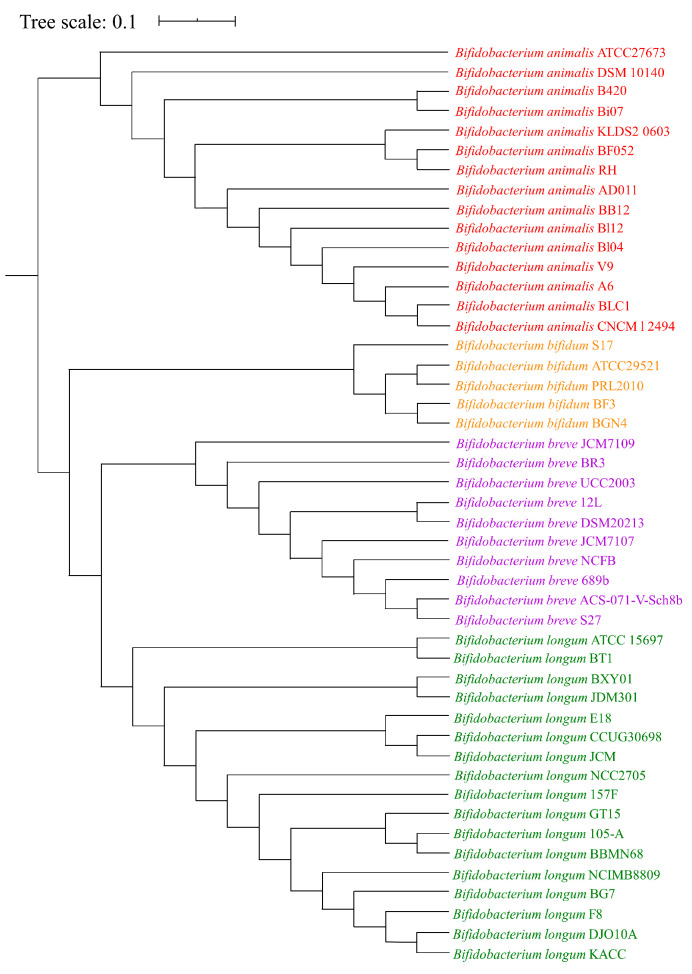
FGD tree of 47 *Bifidobacteium* genomes from four species. The functional relationships among these species were assessed through FGD analysis derived from predicted ORF amino acid sequence similarities. All strains from each species formed a functional cluster.

**Figure 3 foods-14-04243-f003:**
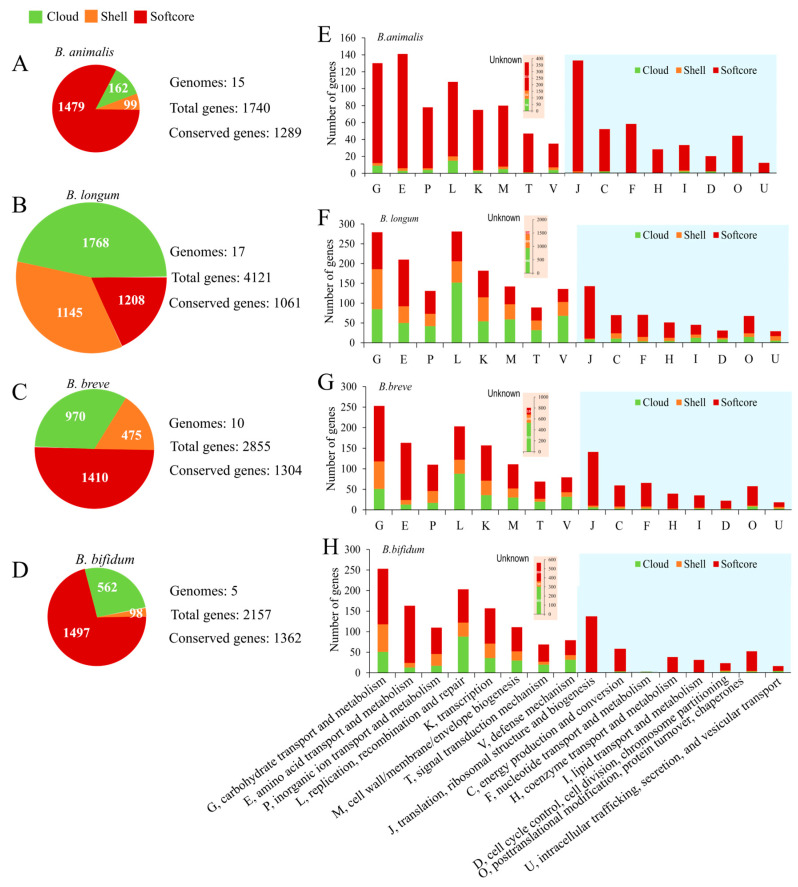
Comparative gene repertoire and functional classification (COG) of four *Bifidobacterium* species. Composition and size of the gene repertoires identified in each species (**A**–**D**). Cloud (present in ≤2 genomes), shell (3 ≤ *n* < N − 1 genomes), and softcore (present in ≥N − 1 genomes). The COG classification of each species (**E**–**H**). The *y* axis represents the number of genes, whereas the *x* axis represents categories of COGs. *B. animalis* (**A**,**E**), *B. longum* (**B**,**F**), *B. breve* (**C**,**G**), *B. bifidum* (**D**,**H**).

**Figure 4 foods-14-04243-f004:**
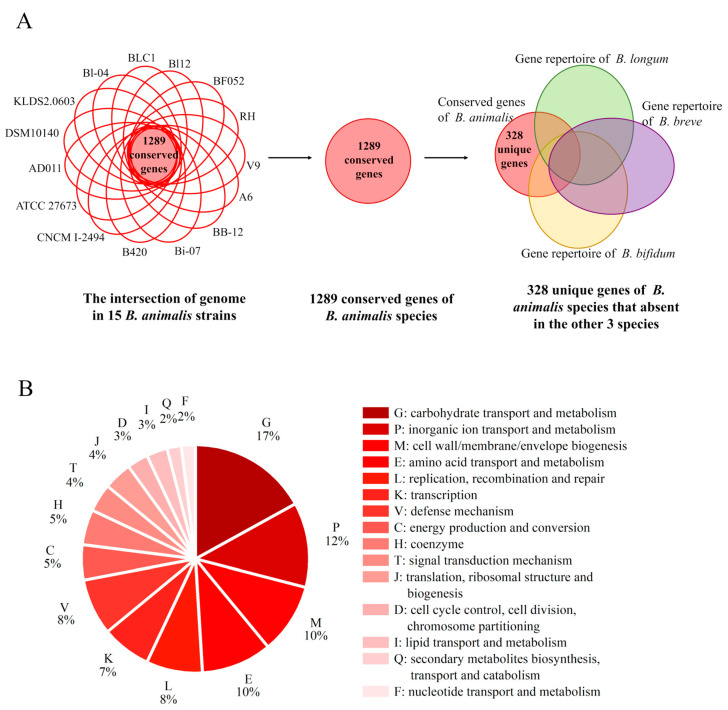
The screening process and COG classification of the truly unique genes in *B. animalis*. (**A**) The screening process of the 328 truly unique genes in *B. animalis*. (**B**) COG classification of the 148 annotated truly unique genes in *B. animalis*.

**Figure 5 foods-14-04243-f005:**
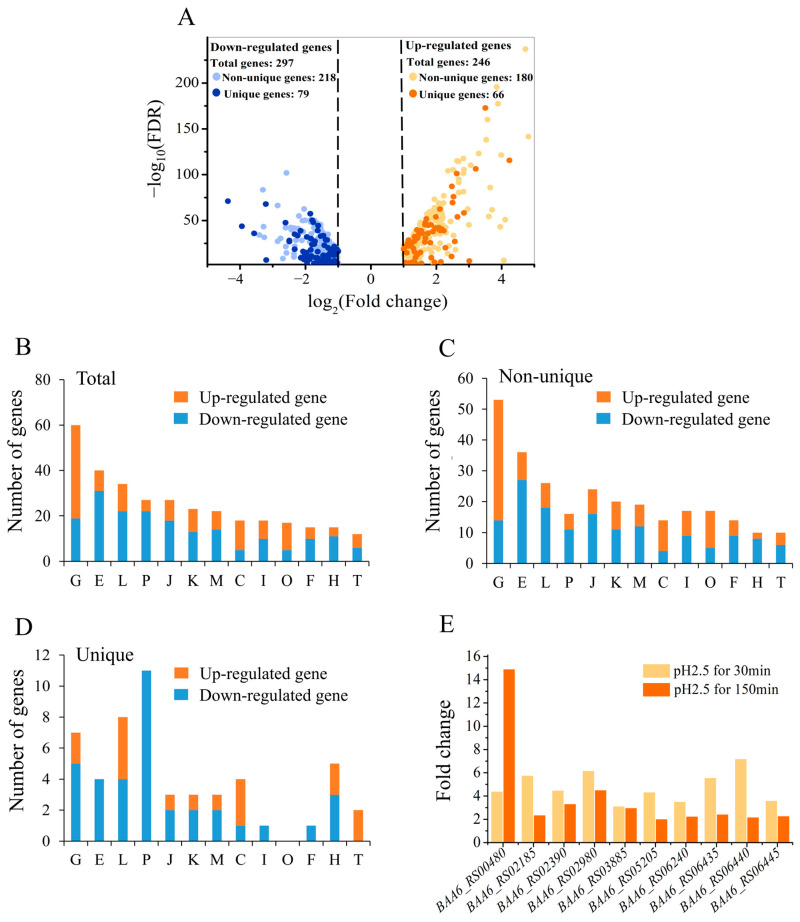
Transcriptome analysis of the *B. animalis* A6 strain under pH 2.5 for 30 min. Volcano plot of differentially expressed genes (DEGs) in the A6 strain under pH 2.5 for 30 min, compared with the non-treated cells (**A**). The putative functions of the total DEGs (**B**), non-unique DEGs (**C**) and unique DEGs (**D**) were classified into different categories by COGs. The y axis represents the number of genes, whereas the x axis represents the category of COGs, with detailed descriptions for each category provided in (**B**). Fold-change of expression of the 10 selected genes after acid treatment (**E**). The result of the pH 2.5 for 30 min group was obtained by RNA sequencing, while the result of the pH 2.5 for 150 min group was obtained by qRT-PCR.

**Figure 6 foods-14-04243-f006:**
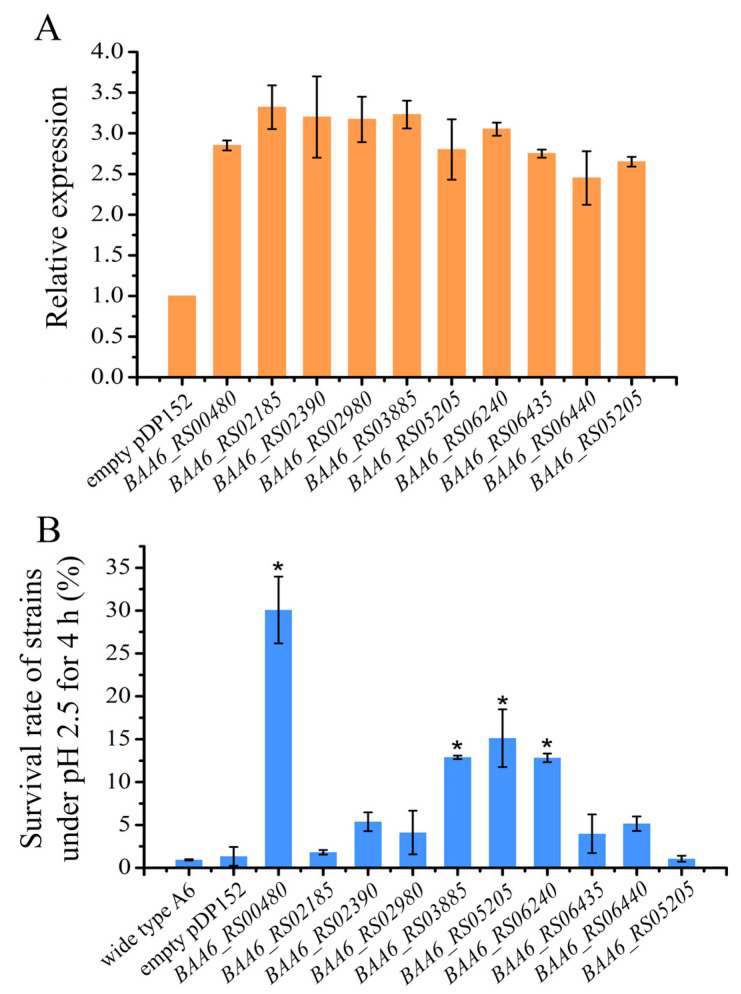
Relative expression of genes and survival rate of the overexpression strains. (**A**), relative expression of each overexpression gene. (**B**) The survival rate of each overexpression strain under pH 2.5 for 4 h. “*” indicates that there is a significant difference between overexpression strains and control strain (*p* < 0.05).

## Data Availability

All raw sequencing data have been deposited into the NCBI Sequence Read Archive (SRA) database under the accession numbers PRJNA431956.

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
