# Peer review of "Comparative Genomics Analysis Combined with Homologous Overexpression Reveals the Mechanism of Species-Specific Acid Stress Resistance in Bifidobacterium animalis"

_foods, 2025, doi:10.3390/foods14244243_

Round 1

Reviewer 1 Report

Comments and Suggestions for Authors

The manuscrit is about comparative genomics analysis combined with homologous overexpression reveals the mechanism of species-specific acid stress résistance in Bifidobacterium animalis 

The manuscrit is well written and presented, the results are very interesting and needs to be developed in vivo experimental design. 

Howerver, I have some remarks 

Introduction 

The aim of the Study should be clearly defined 

Materials and Methods 

Line 83-84 : The Origin of the fecal Material should be specified 

Line 96-103: add reference 

Line 133-139 : add reference 

Line 168-205 : add reference 

The authors should add a conclusion and perspectives of the Study 

Reviewer 2 Report

Comments and Suggestions for Authors

please find my comments in my review report

Reviewer 3 Report

Comments and Suggestions for Authors
  1. Previous studies—such as those by Matsumoto et al. and Sanchez et al.—have already explored the acid tolerance of animalis. The authors are kindly requested to clearly outline the novelty and significance of the present work. In particular, it would be helpful to explicitly state what new aspects this study addresses or extends beyond the findings of the earlier literature.
  2. The studies mentioned in lines 48-53 should clearly provide the quantitative comparisons rather than simply stating the superiority of the animalis compared to other species
  3. Use the full form of the abbreviation for the first time in the manuscript (RESM: Line 154)
  4. Avoid using person-centred statements in the manuscript (Line 387, 396, 428 etc.). Look for these statements throughout the study and make necessary changes
  5. It appears that the manuscript still contains template instructions after the Conclusion section (lines 435–437: “Authors should discuss the results and how they can be interpreted…”). These are part of the journal’s formatting guidelines and should not be included in the main text. I recommend removing these instructional lines to maintain clarity and ensure the manuscript adheres to the journal’s formatting standards.
  6. As the author states, the manuscript only studied the differential gene expression of animalis under acid stress conditions. Still, it lacks the mechanism of action for these genes in inculcating this resistance. It should be included in the limitations of this study
  7. The references of the manuscript should be reported following the journal guidelines.
  8. Language can be improved by avoiding the use of long sentences, which will increase the flow and scientific tone of the manuscript.
  9. The authors are encouraged to elaborate on the potential future directions of this work and discuss how the findings may be applied in food or probiotic-related applications. Highlighting these practical implications would help strengthen the overall impact of the study.
